# A Novel Hybrid Reactor of Pressure-Retarded Osmosis Coupling with Activated Sludge Process for Simultaneously Treating Concentrated Seawater Brine and Wastewater and Recovering Energy

**DOI:** 10.3390/membranes12040380

**Published:** 2022-03-31

**Authors:** Jiaxin Ding, Qian Zhou, Zixun Zhou, Wenyu Chu, Yao Jiang, Wei Lai, Pin Zhao, Xinhua Wang

**Affiliations:** 1Jiangsu Key Laboratory of Anaerobic Biotechnology, School of Environmental and Civil Engineering, Jiangnan University, Wuxi 214122, China; dingjiaxin1003@163.com (J.D.); zqzqzhouqian@163.com (Q.Z.); zhouzixunjnu@163.com (Z.Z.); aa80231000@foxmail.com (W.C.); yookang2022@163.com (Y.J.); laiwei1124@163.com (W.L.); 2Jiangsu Collaborative Innovation Center of Technology and Material of Water Treatment, Suzhou University of Science and Technology, Suzhou 215009, China

**Keywords:** seawater brine, municipal wastewater, pressure retarded osmosis, energy recovery, water reclamation

## Abstract

As an attractive way to deal with fresh water shortage, membrane-based desalination technologies are receiving increased interest. However, concentrated seawater brine, in needing further treatment, remains a main obstacle for desalination via membrane technology. Here, a hybrid technology integrating pressure-retarded osmosis with activated sludge process (PRO-MBR) was applied for simultaneously treating concentrated seawater brine and municipal wastewater. Performance of the PRO-MBR, including water flux, power density, contaminants removal, and membrane fouling was evaluated and compared at two different membrane orientations (i.e., active layer facing feed solution (AL-FS) mode and active layer facing draw solution (AL-DS) mode). During the PRO-MBR process, the municipal wastewater was completely treated regardless of the membrane orientation, which means that there was no concentrated sewage needing further treatment, owing to the biodegradation of microorganisms in the bioreactor. In the meantime, the concentrated brine of seawater desalination was diluted into the salinity level of seawater, which met the standard of seawater discharge. Owing to the high rejection of forward osmosis (FO) membrane, the removal efficiency of total organic carbon (TOC), total phosphorus (TP), ammonia nitrogen (NH_4_^+^-N), and total nitrogen (TN) was higher than 90% at both modes in the PRO-MBR. In addition, the PRO-MBR can simultaneously recover the existing osmotic energy between the municipal wastewater and the seawater brine at both modes. Compared with the AL-DS mode, the AL-FS mode took a shorter time and achieved a bigger power density to reach the same terminal point of the PRO-MBR owing to a better water flux performance. Furthermore, the membrane fouling was much more severe in the AL-DS mode. In conclusion, the current study demonstrated that the PRO-MBR at the AL-FS mode can be a promising and sustainable brine concentrate and municipal wastewater treatment technology for its simultaneous energy and water recovery.

## 1. Introduction

Recently, freshwater scarcity is becoming more and more serious along with climate change and population growth, and is one of the most pressing challenges for up to four billion people. Desalination is an attractive technology to deal with freshwater shortage in many regions of the world, due to the abundance of seawater [1,2]. Over the past several decades, reverse osmosis (RO) has dominated the market of seawater desalination due to its ease of operation, simple design, and maintenance [3,4,5]. Moreover, the energy consumption of RO in seawater desalination has decreased from 20 kWh·m^−3^ in the 1970s to about 2 kWh·m^−3^ at 50% recovery now [6,7]. Even though RO is considered the gold standard for seawater desalination, the development of RO in desalination is limited by the disposal of RO concentrate [8]. RO concentrate of seawater desalination is difficult to treat because of the high concentration of salts and organics. The most common disposal solution of RO brine is discharged to the sea, which can diffuse into the marine medium quickly. However, it is increasingly acknowledged that the direct disposal of uncontrolled RO concentrate will increase the total dissolved solid concentration of seawater and disrupt the balance of marine ecosystems [9]. It is of global concern to address RO concentrate of seawater properly. Recently, many technologies, such as evaporation, distillation, electrodialysis, or an integrated membrane operation have been studied in RO concentrate treatment [10,11,12]. Among them, thermal-based methods are energy-intensive, which are not suitable for large flow rates; while membrane-based technologies show great potential in RO concentrate due to their low energy consumption and high efficiency [13,14,15].

As an osmotically driven membrane process, pressure-retarded osmosis (PRO) can recover energy by harnessing the osmotic pressure difference across the membrane of the feed and draw stream. It is well-known that the mixing of dilute and concentrated solutions will release energy. In PRO, the draw stream is diluted, attributed to the increased permeation from feed to draw stream. By applying a hydrostatic pressure which is less than the osmotic pressure to the draw side, the permeation from feed to draw stream works against the hydrostatic pressure and generates power. Consequently, the PRO process dilutes the draw solution (DS) and concentrates the feed solution (FS) prior to discharge, and at the same time, effectively converts the chemical potential from the osmotic gradient into mechanical energy [16]. Compared with the traditional energy production process, PRO technology is much cleaner and environmentally friendlier [17]. It is very popular to use concentrated brine from seawater desalination and municipal wastewater as feed pairs for PRO, since that RO concentrate is diluted and then discharged to the sea, and in the meanwhile, municipal wastewater is concentrated [18,19]. However, concentrated municipal wastewater with an accumulation of contaminants needs further treatment for discharging or reuse, which becomes a drawback hindering the application of PRO in concentrated brine during seawater desalination.

Recently, Meng et al. proposed a novel concept of PRO-membrane bioreactor (PRO-MBR), which combines the PRO and the activated sludge process [20]. The PRO-MBR was used to treat the municipal wastewater for simultaneously achieving the water reclamation and energy recovery. The results indicated that excellent contaminants removal was achieved owing to the biodegradation of microorganisms and the rejection of forward osmosis (FO) membrane, and the contaminants build-up in the municipal wastewater was effectively alleviated. In addition, the power density of the PRO-MBR ranged from 1.67 to 3.34 W·m^−2^. However, membrane fouling of the PRO-MBR was severe, owing to the porous support layer of the FO membrane facing the activated sludge. It should be pointed out that there is only one paper on the PRO-MBR (using municipal wastewater and NaCl solution as the FS and DS, respectively) under a short-term operation, thus it is necessary to start more research, especially over longer operation times, for better understanding the PRO-MBR.

Inspired by the success of PRO-MBR for treating municipal wastewater, we intend to apply the PRO-MBR process for simultaneous recovery of water and energy in this study. Specifically, concentrated brine from seawater desalination and municipal wastewater were used as the DS and FS, respectively. After the treatment, the brine from seawater desalination was diluted to the salinity level of ocean water and could be directly discharged to the sea. At the same time, the municipal wastewater was concentrated and treated by the PRO-MBR reactor. Correspondingly, the aim of the current study was to investigate the performance and membrane fouling of the PRO-MBR in the brine of seawater desalination dilution and municipal wastewater treatment. To the authors’ knowledge, this is the first study on the application of the PRO-MBR in simultaneously treating the concentrated brine of seawater desalination and municipal wastewater.

## 2. Materials and Methods

### 2.1. Feed and Draw Solutions

Simulated domestic sewage and RO concentrate of seawater were prepared and applied as the FS and DS in our work, respectively [21]. Simulated domestic sewage contains carbon, nitrogen, phosphorus, and other nutrients; the specific components are shown in Table 1. It is worth mentioning that real domestic wastewater contains lots of other contaminants such as phosphates or silica, which will also affect membrane fouling. These substances are not considered in this study due to their small amounts. On the other hand, the specific components of inorganic salts in seawater are shown in Table 2.

### 2.2. Experimental Set-Up and Operating Conditions

In this study, experiments were conducted in a bench-scale PRO-MBR setup (Figure 1), referring to our previous work [20]. It consisted of a bioreactor, a PRO module, an FS loop, and a DS loop. The bioreactor had an effective volume of 2 L and an aeration diffuser of 100 L/h at the bottom with mixed liquor suspended solids (MLSS) at 3–5 g/L. The PRO module contained two identical channels, holding a flat-sheet membrane with an effective membrane area of 25.5 cm^2^. The mixture of activated sludge (2 L) and simulated municipal sewage (800 mL) in the bioreactor was recirculated through the FS flow channel with a cross-flow velocity of 10 cm/s by a peristaltic pump (BT100-2J, Longer Precision Pump, Baoding, Hebei, China), and the simulated RO concentrate of seawater in the DS tank (800 mL) was recirculated through the DS flow channel with a cross-flow velocity of approximately 168 cm·s^−1^ at a pressure of 4 bars, provided by a high-pressure pump (DP-130, Xinxishan, Shanghai, China). A tricot type spacer was selected to fill the draw side of PRO cell to enhance the mechanical strength of membrane. Weight increment of DS was monitored continuously by a weighing balance (PL6001E, Satorius weighing technology GmbH, Gottingen, Germany) for the calculation of water flux. The PRO-MBR process was operated at 25 ± 2 °C with dissolved oxygen (DO) concentration of 1–5 mg/L.

### 2.3. Membrane Properties and Orientation

The membrane used in this study was a commercial flat-sheet cellulose triacetate (CTA) FO membrane, which was provided by Hydration Technology Innovations (HTI), Albany, NY, USA. The CTA membrane shares general characteristics of asymmetric structure, which has two different layers: active layer (AL) made from cellulose acetate and support layer (polyester woven mesh, SL). Its water permeability coefficients (A) and salt permeability coefficients (B) were approximately 0.8 L/(m^2^·h·bar) and 2 × 10^−7^ m/s, respectively [22].

In the PRO-MBR, there are two modes of membrane orientation due to the different properties of two layers. Specifically, one is AL-FS mode, in which dense AL facing FS and porous SL facing DS; and the other one is AL-DS mode with SL facing FS and AL facing DS.

### 2.4. Analytical Methods

The water flux, Jw, given in units of liters per square meter per hour (LMH), was calculated by recording the mass change in the feed solution reservoir versus time.
(1)$$J_{w}=\frac{\Delta m}{\rho \times A\times \Delta t}$$
where Δm (g) is the mass variation over the time interval Δt (h), A is the effective membrane surface area, 25.5 cm^2^, and ρ is the density of water, 1000 g/L.

The power density of the PRO-MBR was defined as the osmotic power per unit membrane area according to previous reports [17,23]:(2)$$W=\frac{J_{w}\times \Delta P}{36}$$
where W is the power density (W/m^2^), J_w_ is the water flux of the FO membrane (LMH), and ΔP is the effective hydraulic pressure difference across the membrane (bars).

Moreover, measurements of mixed liquor suspended solids (MLSS), mixed liquor volatile suspended solids (MLVSS), total solids (TS), volatile solids (VS), chemical oxygen demand (COD), NH_4_^+^-N, total nitrogen (TN), and total phosphorus (TP) were conducted according to the Standard Method (APHA, 1998), and total organic carbon (TOC) concentration was analyzed by a TOC analyzer (TOC-VCPH, Shimadzu, Japan).

After fouling experiments, the fouled FO membranes were taken out carefully from the PRO cell for further analyses. The morphology and element composition of the fouled FO membranes were determined by field emission scanning electron microscope (FE-SEM) (S4800, Hitachi, Japan) and energy diffusive X-ray (EDX) analyzer (Falcon, EDAX Inc., Rochester, NY, USA), respectively. Furthermore, the distributions of organic foulants and biofoulants on fouled membrane were analyzed by using a confocal laser scanning microscope (CLSM, LSM 710, ZESIS, Oberkochen, Germany).

## 3. Results and Discussion

### 3.1. The Performance of PRO-MBR at Two Different Membrane Orientations

The performance aspects of PRO-MBR, including water flux, power density, and contaminants removal, were investigated at both AL-FS and AL-DS membrane orientations. Results for each performance aspect are presented in Figure 2 and Figure 3, and Table 3, respectively.

#### 3.1.1. Water Flux

As shown in Figure 2a, membrane orientation had a vital impact on the permeation performance. Water flux of AL-DS mode achieved an initial value of 22 LMH, declined significantly, and then remained at about 4 LMH. The water flux of AL-FS mode was about 14 LMH at first, and then dropped gradually at the end of 8 LMH. In order to further evaluate the tread of water flux at both orientations, the normalized flux as a function of operating time was calculated and is illustrated in Figure 2b. There was a two-stage variation of trans-membrane fluxes with time in the AL-FS mode. In the first stage of 300 min, water flux declined rapidly, which was likely attributed to the declined osmotic pressure in the draw side and the formation of the cake-enhanced osmotic pressure in the surface of membrane [24]. In the second stage from 300 min to the end, the water flux declined steadily. In this stage, the membrane fouling built up gradually, and then led the water permeation to decline. On the other hand, besides the first two stages corresponding to the condition of AL-FS mode, AL-DS mode has another stage, in which the membrane fouling achieved a dynamic equilibrium, and thus the permeate flux kept stable (after 1380 min).

Comparing with the AL-FS mode, the AL-DS mode had a higher water flux in the first 480 min, but subsequently a significant flux drop. Moreover, despite the high initial water flux, AL-DS mode (2100 min) took more time than AL-FS mode (1680 min) to reach the end of the PRO-MBR, i.e., diluting the RO brine to the salinity level of seawater. These results might be attributed to the porous support layer facing the activated sludge directly in the AL-DS mode, resulting in less concentrative internal concentration polarization (ICP) but high membrane fouling tendency [25,26]. Moreover, the high initial water flux of AL-DS mode would exacerbate the membrane fouling [21]. Compared with the previous report on PRO-MBR treating municipal wastewater in the AL-DS mode, a similar initial flux was achieved in the current study. However, owing to a relatively short operating time of 240 min, the previous study just observed the variation of water flux at the first two stages [20]. Comprehensively considering both the flux value and the variation trend, the AL-FS mode was more suitable for the PRO-MBR over long-term operation.

#### 3.1.2. Power Density

In accordance with Equation (2), the power density of the PRO-MBR was calculated based on the water flux and hydrostatic pressure. The power generated during the PRO-MBR process is directly proportional to the water flux of the FO membrane (LMH) and the effective hydraulic pressure difference across the membrane (ΔP, bars). In our work, the ΔP remained consistent, thus, the power was determined by water permeation. The power density had a similar variation trend with water flux of FO membrane. As presented in Figure 3, the value of power density was ranged from 0.7 to 1.7 W/m^2^ in the AL-DS mode, and 0.4–2.5 W/m^2^ in the AL-FS mode. The power density had a similar variation trend with water flux of FO membrane, which was easy to understand. In the beginning, the AL-DS mode produced more energy than the AL-FS mode. However, the power produced in the AL-DS mode declined faster than in the AL-FS mode. In the whole process of 1680 min, the total power produced in the AL-FS mode was larger than in the AL-DS mode.

The low hydraulic pressure difference is a main factor of low power density. In our work, the ΔP remained consistent (6 bars), which is lower than the theoretical optimum (around 45 bars for present study) for power generation, with the aim of preventing membrane deformation under long-term operation. To improve the power generation performance of PRO-MBR, it is vital to fabricate novel FO membranes with high mechanical strength that could withstand high hydraulic pressure. On the other hand, AL-DS mode exhibited the maximum power density of 2.6 W/m^2^ at the very beginning of operation. However, it declined rapidly as the operation proceeded. This can be attributed to the formation of a fouling layer on the support layer of the FO membrane during the initial filtration. If such membrane fouling can be mitigated, the power density and technoeconomic competitiveness of PRO-MBR will be largely improved.

#### 3.1.3. Contaminant Removal

The concentration and retention efficiency of organic matter and nutrients consisting of TOC, TP, NH_4_^+^-N, and TN in both AL-DS and AL-FS modes are summarized in Table 3. For all targeted matter, the removal rates were higher than 90% in the PRO-MBR regardless of the membrane orientation, which was mainly attributed to both the biodegradation of microorganisms in the bioreactor and the high rejection of FO membrane [27,28]. The excellent contaminant removal was consistent with a previous report on PRO-MBR [20]. In addition, the accumulation of TOC and NH_4_^+^-N was not observed in the feed side of PRO-MBR owing to the aerobic degradation of microorganisms, while a slight build-up of TN and TP was found in the activated sludge. In fact, if the bioreactor of the PRO-MBR was coupled with an anoxic zone and started to discharge the waste-activated sludge, the accumulation of TN and TP can be mitigated. In this case, compared with the typical PRO process, the PRO-MBR can effectively alleviate the accumulation of contaminants in the feed side. It is generally believed that the rejection efficiency of AL-DS mode is lower than that of AL-FS mode mainly due to the build-up of contaminants in the porous support layer of feed side in the single FO membrane permeation process [29]. However, there were no significant differences in the contaminant’s removal between the AL-FS and AL-DS mode in the PRO-MBR, which was due to the role of biodegradation in the bioreactor promoting the removal efficiency. In the end, the diluted brine of seawater desalination (i.e., the draw solution of the PRO-MBR) contained 2.34 mg/L of TOC, 0.02 mg/L of TP, 0.16 mg/L of NH_4_^+^-N and 0.48 mg/L of TN in the AL-DS mode, and 1.69 mg/L of TOC, 0.16 mg/L of TP, 0.07 mg/L of NH_4_^+^-N and 0.31 mg/L of TN in the AL-FS mode. The yielded water reached the Chinese national first-class standard for sea water discharging (GB 8978-1996), and could be discharged back into the sea directly.

### 3.2. Membrane Fouling of PRO-MBR at Two Different Membrane Orientations

The fouled membranes were carefully taken out from the PRO module after the operation of PRO-MBR. To investigate the difference of membrane fouling in the two membrane orientations of PRO-MBR process, optical camera and SEM were used to determine the surface morphology. As described in Figure 4a,b, the surface morphology of fouling layer was totally different in the AL-FS and AL-DS mode. It is apparent that the amount of foulants attached to the membrane surface of AL-FS mode was much less than in AL-DS mode. There were large amounts of crystal particles in the fouling layer of the AL-DS mode and very few in the AL-FS mode. Moreover, the amount of TS and VS in both AL-FS and AL-DS modes of PRO-MBR process was calculated for the foulants composition analyses. As described in Table 4, the attached foulants of AL-FS mode (7.61 ± 0.82 g/m^2^) was much less than that of AL-DS mode (17.65 ± 1.57 g/m^2^), which was consistent with the information based on the original photos of fouling membranes. Based on the data of VS/TS, the main kind of foulants in the AL-FS mode was organic matter and biofoulants (approximately 70%), and the content of organic matter and biofoulants was similar with inorganic matter in the AL-DS mode.

Furthermore, EDX and the CLSM coupled with the multiple fluorescence labeling were measured to analyze the typical inorganic, organic and biological foulants in both AL-FS and AL-DS modes. Base on the results of EDX analysis in Figure 4c, there were significant amounts of Ca in the fouling layer of AL-DS mode, while very little Ca in the AL-FS mode. It indicated the presence of severe scaling in the AL-DS mode and less inorganic fouling in the AL-FS mode. The foulants at both AL-FS and AL-DS modes had the indicator elements of organic and biological pollutants, including C, O, N, and P [30,31,32]. In addition, as illustrated in Figure 4d, polysaccharides, proteins, and microorganisms distributed dispersedly in the fouled membrane of both AL-FS and AL-DS modes in different proportions. What’s more, quantitative analysis of structural parameters about the CLSM images were obtained according to Auto PHLIP-ML software (PHLIP, version 1.0) [33,34,35,36], which are summarized in Table 5. It can be seen clearly that the AL-DS mode had more polysaccharides and proteins, less microorganisms, and thicker fouling layer than the AL-DS mode.

Based on the above results, the AL-DS mode suffered more severe membrane fouling than the AL-FS mode, mainly due to the following two factors: 1. the initial water flux of AL-DS was high, which would accelerate the formation of severe membrane fouling; 2. contaminants tended to adhere into the support layer in the AL-DS mode because of the high roughness of the support layer [24,37].

## 4. Conclusions

In this study, the concentrated brine of seawater desalination was efficiently diluted to the salinity level of seawater, and in the meanwhile, municipal wastewater was treated with energy recovery through a novel PRO-MBR process. The performance of PRO-MBR at two different membrane orientations (AL-FS and AL-DS mode) was evaluated in terms of water flux, power density, removal rate of contaminants, and membrane fouling. Water flux of the AL-DS mode was higher in the initial stage, but decreased rapidly. The AL-FS mode (1680 min) was more efficient than the AL-DS mode (2100 min) for diluting the concentrated brine twice. Moreover, removal rate of contaminants was satisfactory in both AL-DS and AL-FS modes. The diluted brine could be disposed into the sea directly. Results of SEM, EDX, and CLSM measurements illustrated that, compared to the AL-FS mode, the AL-DS mode suffered from more severe membrane fouling, which might be attributed to a higher initial water flux and a higher roughness of support layer. In brief, it is feasible to apply the PRO-MBR process especially at the AL-FS mode to simultaneously achieve the dilution of seawater concentrate and the treatment of municipal wastewater with the recovery of energy.

## Figures and Tables

**Figure 1 membranes-12-00380-f001:**
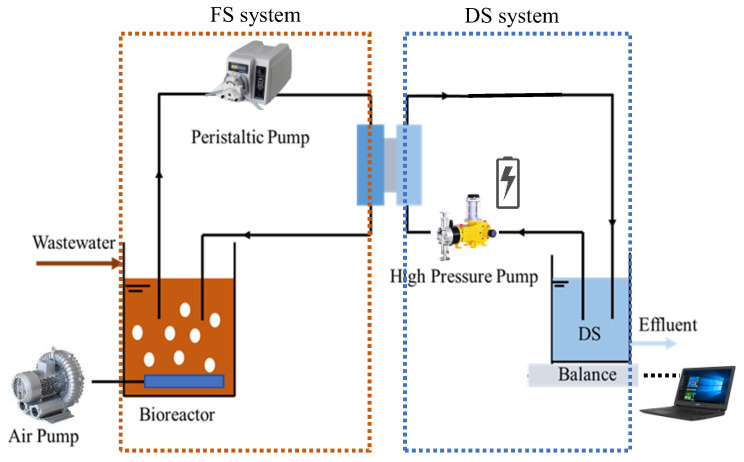
Schematic diagram of the laboratory-scale PRO-MBR system.

**Figure 2 membranes-12-00380-f002:**
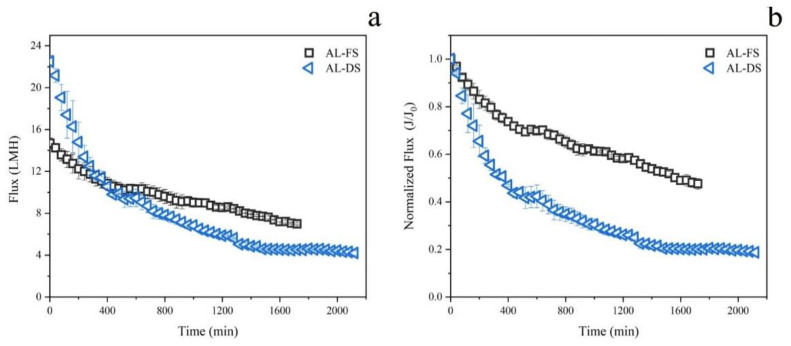
The variation of water flux at AL-FS and AL-DS membrane orientations. (**a**) the absolute value of water flux, (**b**) the normalized water flux. All the data points were determined three times, and their mean values and standard deviation were given.

**Figure 3 membranes-12-00380-f003:**
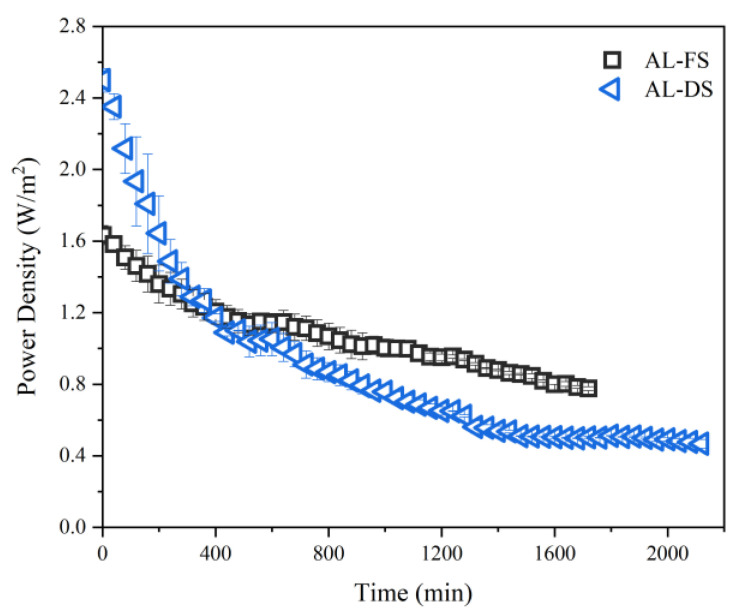
The variation of power density at both AL-FS and AL-DS membrane orientations.

**Figure 4 membranes-12-00380-f004:**
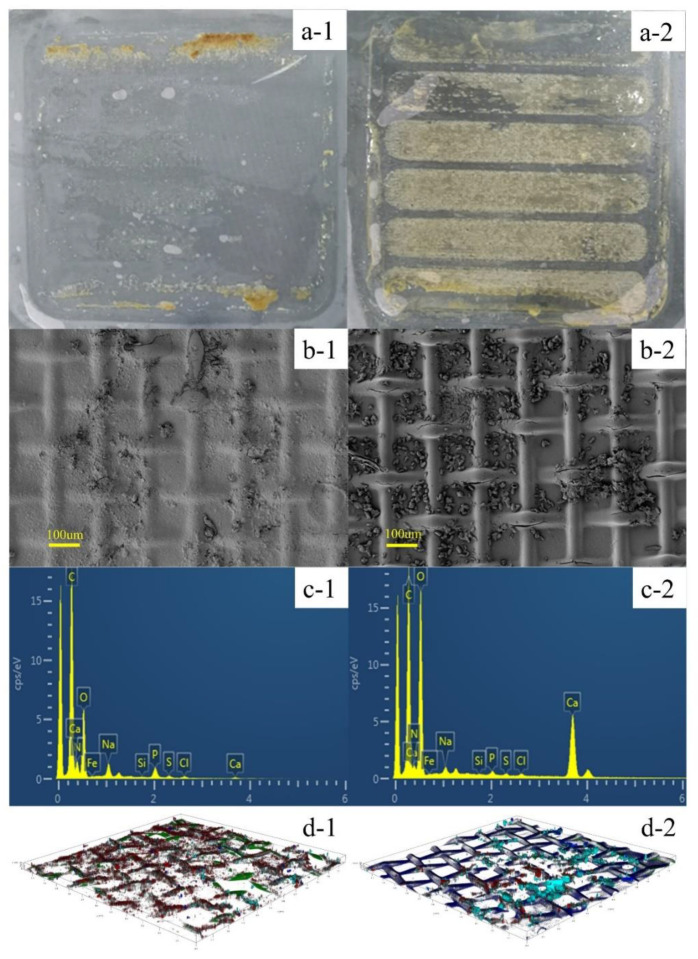
(**a**) The original picture of fouled membrane surfaces; (**b**) SEM and (**c**) EDX image of membrane surfaces after the PRO-MBR operation; (**d**) CLSM images of fouled membrane after the PRO-MBR operation in the (1) AL-FS and (2) AL-DS modes. The fluorescence colors of α-D-glucopyranose polysaccharides, β-D-glucopyranose polysaccharides, proteins and microorganisms were cyan, blue, green and red in CLSM images, respectively. (For interpretation of the references to color in this figure legend, the reader is referred to the web version of this article).

**Table 1 membranes-12-00380-t001:** The specific components of simulated domestic sewage.

Components	Concentration (mg/L)	Components	Concentration (mg/L)
Glucose	230.0	Ammonium bicarbonate	170.0
Beef extract	20.0	Potassium dihydrogen Phosphate	28.0
Fish meal peptone	60.0	Ferric chloride	1.0
Sodium acetate (anhydrous)	40.0	Calcium chloride (anhydrous)	1.2
Sodium bicarbonate	198.0	Magnesium chloride (hexahydrate)	2.4

**Table 2 membranes-12-00380-t002:** The specific components of seawater concentrate.

Components	Concentration (g/L)	Osmotic Pressure (atm)	Percentage (%)
NaCl	49.06	39.65	58.71
MgCl_2_	22.22	15.44	26.59
Na_2_SO_4_	8.18	3.27	9.79
CaCl_2_	2.32	1.36	2.78
KCl	1.39	0.87	1.66
NaHCO_3_	0.40	0.22	0.48
Total	83.57	52.60	100

**Table 3 membranes-12-00380-t003:** TOC, TP, NH_4_^+^-N, and TN concentrations in the influent sludge supernatant and FO permeate and their removal rate by PRO-MBR process ^a^.

		AL-DS	AL-FS
TOC	Influent Concentration (mg/L)	75.3 ± 4.56	75.87 ± 3.73
	Concentration in sludge supernatant (mg/L)	1.35 ± 0.57	1.65 ± 0.85
	Concentration in FO permeate (mg/L)	4.68 ± 2.45	3.38 ± 2.51
	Removal rate (%)	93.78 ± 3.25	95.55 ± 3.31
TP	Influent Concentration (mg/L)	5.75 ± 0.08	5.33 ± 0.09
	Concentration in sludge supernatant (mg/L)	5.79 ± 0.05	5.35 ± 0.10
	Concentration in FO permeate (mg/L)	0.04 ± 0.02	0.07 ± 0.04
	Removal rate (%)	99.30 ± 0.16	98.62 ± 0.79
NH_4_^+^-N	Influent Concentration (mg/L)	27.48 ± 0.13	30.70 ± 0.23
	Concentration in sludge supernatant (mg/L)	0.53 ± 0.04	0.18 ± 0.10
	Concentration in FO permeate (mg/L)	0.32 ± 0.07	0.13 ± 0.02
	Removal rate (%)	98.84 ± 0.27	99.59 ± 0.07
TN	Influent Concentration (mg/L)	30.19 ± 0.20	33.91 ± 0.25
	Concentration in sludge supernatant (mg/L)	52.88 ± 0.89	56.26 ± 0.27
	Concentration in FO permeate (mg/L)	0.97 ± 0.07	0.62 ± 0.28
	Removal rate (%)	96.78 ± 0.03	98.17 ± 0.81

^a^ Values are given as mean values ± standard deviation (number of measurements: *n* = 3).

**Table 4 membranes-12-00380-t004:** Analyses of the foulants on the membrane surfaces at both membrane orientations ^a^.

Membrane Orientation	TS (g/m^2^)	VS (g/m^2^)	VS/TS (%)
AL-FS	7.61 ± 0.82	5.31 ± 0.47	69.74 ± 1.28
AL-DS	17.65 ± 1.57	9.35 ± 2.03	52.93 ± 6.84

^a^ Values are given as mean values ± standard deviation (number of measurements: *n* = 3).

**Table 5 membranes-12-00380-t005:** Biovolume of the foulants on membrane surfaces in both AL-FS and AL-DS modes after the PRO-MBR operation ^a^.

	α-D-glucopyranose Polysaccharides (μm^3^/μm^2^)	β-D-glucopyranose Polysaccharides (μm^3^/μm^2^)	Protein (μm^3^/μm^2^)	Total Cells (μm^3^/μm^2^)	Thickness (μm)
AL-FS	1.17 ± 0.10	2.70 ± 0.38	0.93 ± 0.06	2.92 ± 0.32	34.69 ± 1.33
AL-DS	1.01 ± 0.07	4.13 ± 0.01	1.18 ± 0.03	0.76 ± 0.15	39.11 ± 0.49

^a^ Values are given as mean values ± standard deviation (number of measurements: *n* = 3).

## Data Availability

Not applicable.

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
