# Peer review of "A Novel Hybrid Reactor of Pressure-Retarded Osmosis Coupling with Activated Sludge Process for Simultaneously Treating Concentrated Seawater Brine and Wastewater and Recovering Energy"

_membranes, 2022, doi:10.3390/membranes12040380_

Round 1
Reviewer 1 Report
The purpose of the manuscript is to evaluate the simultaneous contaminant removal and osmotic energy harvesting from a domestic wastewater (feed) and SWRO retentate (draw) using PRO membranes. The tests were conducted in AL-DS and AL-FS mode to see how the membrane flux and fouling factors are influences. Overall, this is a new attempt at combining two different methodologies to extract maximum value from two waste streams. Even though PRO has been validated at different levels using waste streams from brackish water RO reject and SWRO reject, combining it with an activated sludge bioreactor process certainly presents a new perspective. As such, the experimental design, data interpretation and conclusions drawn, are all in good agreement with each other, authors are requested to review the formatting/typographical/grammatical errors again. Authors are requested to note that even though, the results showcased in this manuscript give us a good preliminary understanding of such a hybrid process, there are several factors that may add more complexity to the system. For e.g. real domestic waste water will have lot of other contaminants such as phosphates or silica, which also foul PRO membranes. Also, authors are requested to note that PRO membranes are being developed fast in terms of performance factors like flux and power density. Some literature examples of 3-5 LMH/bar and up to 20 W/m2 power density have been reported. Therefore, it needs to be highlighted that the PRO membranes in the current study have lower power density compared to literature values, the concept presented here is applicable to PRO process in general and can be adaptable to advanced PRO membranes as well. Please also add a note about what the future research will be, as continuation of this work.
I recommend that this manuscript be accepted in the current form.
Reviewer 2 Report
Overall the paper is interesting as it presents a combination of a biological and membrane process that could enable energy and water recovery from sea water and municipal wastewater.
Some modifications and English improvements should be made.
Abstract
More specific results should be included.
Lines 43 and 44
3 should be superscripted.
Even through should be Even though..
Line 72
The word wastewater is missing after municipal.
Line 73
reusing should reuse.
Line 76
combining should be combines
Line 86
researches should be research.
Line 88
It should be ... Inspired by the success..
Line 104
It should be.. On the other hand, the specific components of inorganic salts in seawater are shown in Table 2.
Line 115
..The PRO module contained two identical channels, holding the flat-sheet membrane with an effective membrane area of ..
Line 120
1 should be superscripted.
Line 121
Bars should be in metric units throughout the paper.
Line 150
2 should be superscripted.
Line 164
Discussions should be Discussion.
Line 177
Attribute should be attributed.
Line 180
lead should be led.
Line 194
..in the current study
Section 3.1.2
Explanation as to why the power from AL-FS mode was higher than AL-DS mode. In addition, some discussion regarding the power achieved should be added. It is not very high and thus would not be feasible.
Line 213
It should be Contaminant removal.
Line 214
It should be organic matter.
Line 219
The excellent contaminant removal...
Line 260
...were plenty of Ca in the fouling layer.. should be there were significant amounts of Ca in the fouling layer..
Line 268
Information about PHLIP software should be included in the methodology with supplier and version.
Line 275
...the support layer
Line 269
... seen clearly that the...
Figure 4
The scales of the SEM images should be included. The identification of the elements is not clear in c-1 and c-2.
